# Radio Frequency Properties of a 3D Printed Klystron Circuit

**Charlotte Wehner** [1,2,*] **, Bradley Shirley** [2] **, Garrett Mathesen** [2] **, Julian Merrick** [2] **, Brandon Weatherford** [2] **and Emilio Alessandro Nanni** [2]

1   Department of Electrical Engineering, Stanford University, 350 Jane Stanford Way, Stanford, CA 94305, USA
2   SLAC National Accelerator Laboratory, 2575 Sand Hill Road, Menlo Park, CA 94025, USA; brads@slac.stanford.edu (B.S.); mathesen@slac.stanford.edu (G.M.); julianm@slac.stanford.edu (J.M.); brweathe@slac.stanford.edu (B.W.); nanni@slac.stanford.edu (E.A.N.)
*   Correspondence: cwehner@stanford.edu

**Abstract:** The manufacturing of active RF devices like klystrons is dominated by expensive and time-consuming cycles of machining and brazing. In this article, we characterize the RF properties of X-band klystron cavities and an integrated circuit manufactured with a novel additive manufacturing process. Parts are 3D printed in 316 L stainless steel with direct metal laser sintering, electroplated in copper, and brazed in one simple braze cycle. Stand-alone test cavities and integrated circuit cavities were measured throughout the manufacturing process. The un-tuned cavity frequency varies by less than 5% of the intended frequency, and Q factors reach above 1200. A tuning study was performed, and unoptimized tuning pins achieved a tuning range of 138 MHz without compromising Q. Klystron system performance was simulated with as-built cavity parameters and realistic tuning. Together, these results show promise that this process can be used to cheaply and quickly manufacture a new generation of highly integrated high power vacuum devices.

**Keywords:** 3D printing; additive manufacturing; direct metal laser sintering (DMLS); klystron; X-band





## 1. Introduction

Additive manufacturing (AM) continues to see growing interest for the manufacturing and development of RF structures and components. Once confined to prototyping, this family of manufacturing techniques is increasingly used to produce devices previously impossible or impractical to manufacture [1].

One such RF device is a klystron, a linear beam vacuum electron device used to amplify RF signals. Klystrons are typically used as satellite, radar, and communications transmitters, as well as RF power generators to drive particle accelerators [2]. The basis of this work uses the klystron circuit shown in Figure 1, which is a compact four-cavity X-band klystron circuit intended to produce over 300 kW of pulsed RF to power the cavities of a linear accelerator.

The basic operation of a klystron is the conversion of the kinetic energy in a velocity-modulated electron beam to potential energy of the fields induced in a resonant cavity. More specifically for the device used in this paper, a DC electron beam is emitted with 10 A of current by a thermionic cathode, which is then accelerated by a 60 kV potential. This DC beam traverses the length of the circuit and passes through four resonant cavities. The first resonant cavity is fed with an RF signal at the given X-band design frequency of 11.4–11.5 GHz, establishing an alternating voltage across the gap of the cavity. This alternating electric field does work on the passing electrons to accelerate or decelerate them, depending on which half of the RF period the electrons pass through. This action initiates the bunching of the electron beam at the operating frequency.

A common figure of merit to quantify bunching in the electron beam, and thus RF power extraction efficiency, is to analyze the harmonic current in the beam at a given axial

location. The first few harmonic current values are shown in Figure 2 along the length of the tube. These data were extracted using CERN's KlyC code, a 1.5-D large-signal simulation code for klystrons [3]. It can be seen that the first cavity initiates the bunching of the beam shown as an increase in the harmonic current. The exact distance of the drift space between subsequent cavities is critical to allow slower electrons to catch up to the faster electrons before the next resonant cavity bunches the beam further. Finally, the fully bunched beam reaches the final cavity, which is capacitively tuned (i.e., tuned lower than the fundamental harmonic frequency), causing each bunch to decelerate and convert most of its kinetic energy into potential energy in the form of resonant fields in the cavity which can be extracted by an external waveguide.

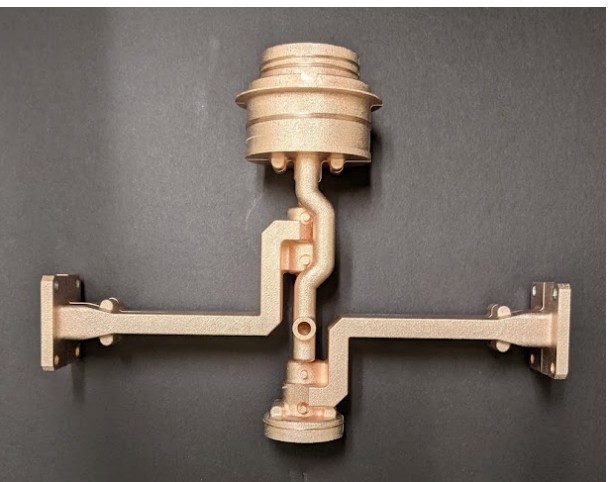

**Figure 1.** Complete klystron circuit with input and output waveguides, pump-out tubes, mounting for an electron gun and collector, and cavity tuning pins.

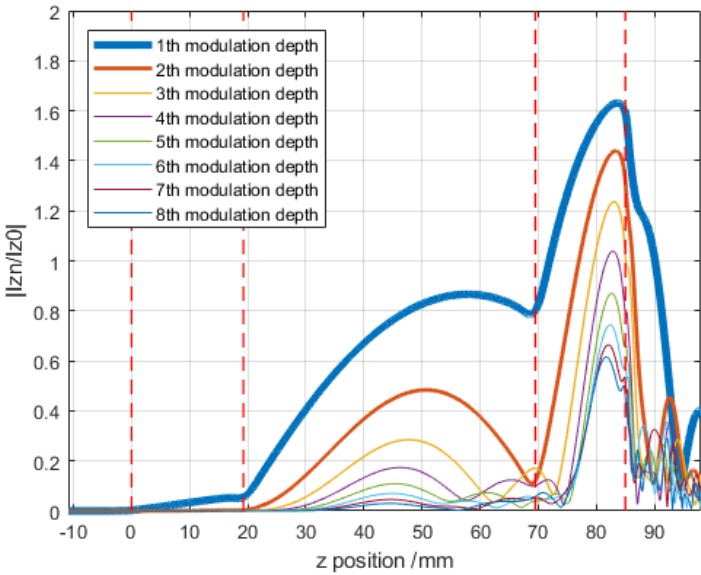

**Figure 2.** First eight harmonic current values normalized to the average beam current as a function of axial distance. Vertical red dashed lines indicate the position of each resonant cavity.

Many different 3D printing technologies are being explored for RF applications. Common technologies fall into four categories: Fused Filament Fabrication (FFF), Stereolithography (SLA), binder jetting or Direct Metal Laser Sintering (DMLS).

Fused Filament Fabrication (FFF) involves extruding a thermoplastic through a heated nozzle. The nozzle is precisely moved though the build volume to deposit rapidly solidifying plastic onto the previous layer. While invaluable for rapid prototyping or small-scale

production due to the very low cost of many FFF 3D printers, the parts this process produces are usually weaker and limited to specific thermoplastics [4]. FFF has been used to manufacture a C-band microwave isolator with a non-printed ferrite bead [5]. However, the limited material selection and poor resolution and accuracy largely prevent FFF from use in the manufacture of RF components.

Stereolithography (SLA), sometimes known as vat polymerization, uses an LCD and backlight or laser with a galvanometer to selectively expose a thin layer of photosensitive resin. The part is withdrawn from the vat of photoresist as the next layer is exposed through the bottom of the vat. While also limited to certain plastics, this technology produces strong, accurate, and high-resolution parts while remaining very cost effective [6]. Due to these advantages, significant previous work has demonstrated manufacturing RF devices with SLA. Metal-plated resin waveguides and antennas have been rigorously studied and demonstrate promise for higher frequency devices [7,8].

Binder jetting uses a liquid binder to selectively bind together powdered material. Binder is dispensed onto the powder bed by an inkjet to form a 2D layer. Another layer of powder is then deposited. Binder jetting can use a wide variety of powdered materials like metals, ceramics, or polymers. Usually, the binder material is a polymer resin. After printing, the parts often must be post-processed, usually involving baking out the binder and sintering [9]. Binder jetting can produce large, cost effective parts, but so far little work has been done using binder jetting for RF components.

Direct Metal Laser Sintering (DMLS) involves using a high powered laser to selectively sinter metal powder. Many similar technologies like selective laser sintering (SLS) and selective laser melting (SLM) exist. A schematic is presented in Figure 3. A laser is directed across the surface of a bed of metal powder. This selectively sinters the powder, forming a layer and adhering it to the previously sintered layer below. The part is then lowered and a roller deposits another layer of powder onto the bed from the hopper. This repeats until complete, then excess powder must be removed from the part. Often, post-processing techniques like hot isostatic pressing or bead-blasting are performed to improve the mechanical properties and surface quality [10]. Since DMLS can produce metal parts with greater than 99% density in a variety of materials and alloys [11], it has seen significant use for both prototyping and full-scale production in many industries, including for RF components. For example, waveguides, filters, Tees, antennas, and loads have been demonstrated across many frequencies [7,12–15]. Additive manufacturing is also of interest for future large-scale accelerator facilities [16–18].

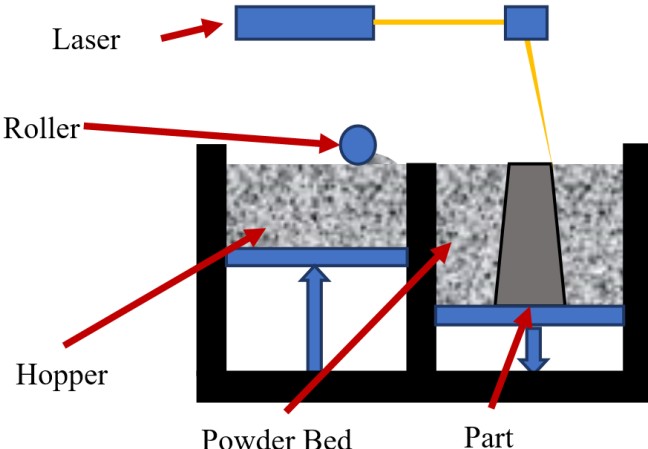

**Figure 3.** Diagram of DMLS printer.

In this article, we present the motivation and process for manufacturing a klystron circuit using DMLS. We also characterize the RF properties of cavities produced with this method. A complete klystron circuit (shown in Figure 1) and 20 test cavities were

manufactured using this process. The RF properties of the test cavities and klystron circuit cavities were measured at multiple stages of the manufacturing process and compared to simulated properties. Also, a tuning study was conducted, and the klystron system performance was simulated with realistic cavity tunings.

## 2. Motivation

A diagram depicting the conventional manufacturing process of a klystron circuit is shown in Figure 4. Due to the limitations of conventional subtractive manufacturing techniques like CNC machining, complex structures like reentrant cavities must be split into smaller, simpler parts. These parts must be machined, polished, and cleaned separately before being brazed into subassemblies. These subassemblies may require additional machining and post processing steps. Next, larger subassemblies are brazed together, followed by additional brazing stages to add additional components like supports, tuning pins, and pumpout tubes. Complex assemblies require many stages of machining and brazing, requiring many hours with skilled technicians and effective coordination between manufacturing departments. This labor-intensive process contributes significantly to the high manufacturing cost of such an RF system [19]. Additionally, every braze joint added to the circuit presents an opportunity for vacuum leaks or water leaks from cooling systems. By offloading the complexity of manufacturing to automated 3D printing systems, the effective complexity and therefore cost of a system could be significantly reduced. Understanding the RF properties of AM cavities and the properties' variations is vital to validating this process.

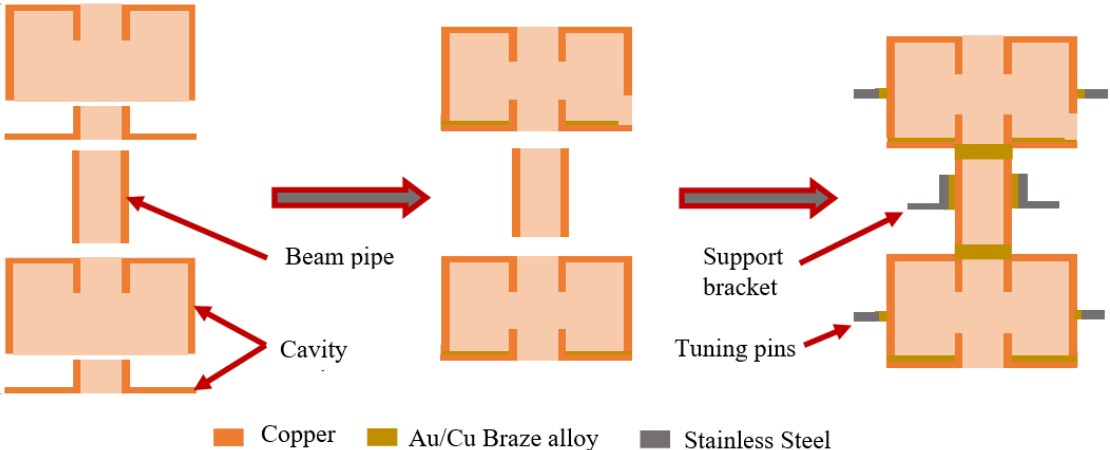

**Figure 4.** Diagram of conventional klystron circuit manufacturing process.

Precision is crucial when manufacturing a klystron. The drift space between each cavity will affect how the electron beam will behave and thus affect the performance of the device. In addition, the frequency of each cavity is directly proportional to its geometry, and any deviations caused by the fabrication process will alter the beam dynamics. Traditionally, the tolerances in the geometries of the cavities are accounted for with the addition of tuning pins to physically alter the geometry of the cavity. With the use of 3D printing, thin cavity walls and optimized tuning geometries can be easily fabricated to enable larger tuning ranges than achievable with conventional manufacturing

## 3. Manufactured Device and Process

The proposed manufacturing process leverages the ability of 3D printing to produce extremely detailed, complex parts that would be either completely impossible or prohibitively expensive with conventional methods. The process for a single cavity is depicted in Figure 5. In DMLS, the cost is driven by the volume of printed material as well as the overall dimensions of the part. Provided that the designer understands the geometric limitations of DMLS, like overhangs and curling, complexity is no longer a limitation. This

allows many cavities and other structures to be combined into only two parts. Printing the entire structure as a single part is impractical due to a reduction of the surface quality of overhanging regions [20] and shadowing effects during electropolishing and plating.

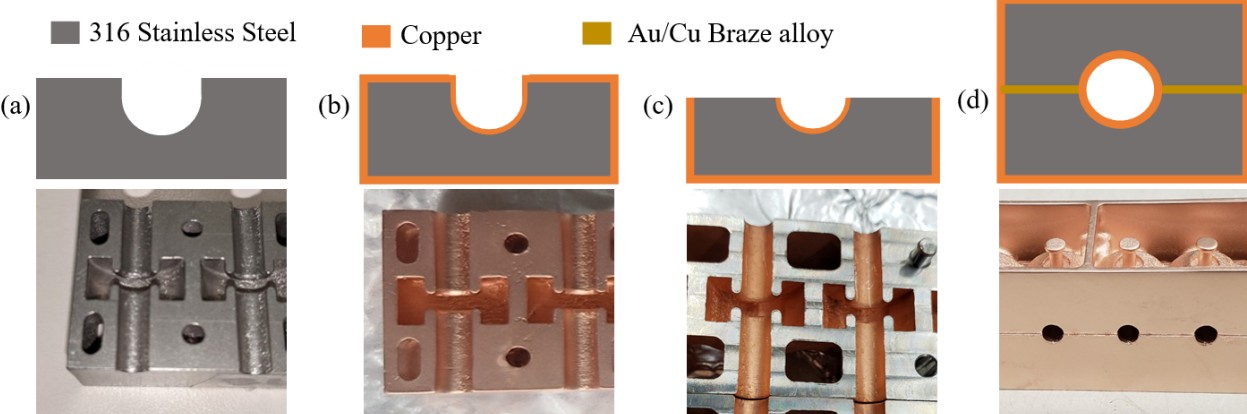

**Figure 5.** Diagram of proposed manufacturing process. (**a**) Printing, (**b**) Plating, (**c**) Face Machining, (**d**) Brazing.

Presently, commercially available DMLS materials do not have favorable properties for directly printing RF cavities [21]. To achieve desirable material electrical properties, the parts are printed in 316 stainless steel, then electroplated with copper. DMLS exhibits significantly higher surface roughness than conventional milling, so the parts are electropolished before Cu plating following standard procedure. Electropolishing reduces small-scale roughness, but not the larger-scale roughness common in DMLS parts. Prior to brazing, the Cu plating was not less than 5 µm, significantly exceeding the 0.6 µm skin depth at this frequency. Figure 6 shows how the plating diffuses into the bulk material during the brazing cycle. It is possible that this raises the resistivity of the material in the skin depth, but this affect was not measured.

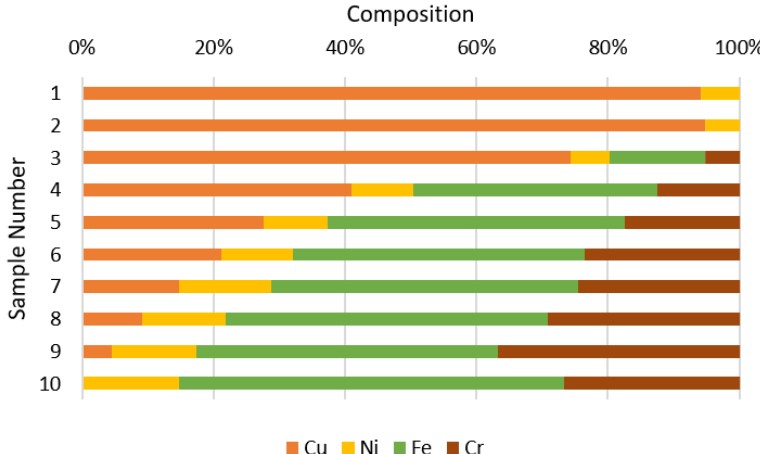

**Figure 6.** Material composition of plated surface measured after brazing with analysis at increasing depths via laser ablation.

The native surface of the DMLS parts is too rough for brazing with thin, laser-cut shims, so the braze surfaces are first faced flat via conventional milling. Though this is a manual machining process, it consists of only one simple operation per part. Laser-cut 0.002″ thick shims of a 25%/75% Au/Cu braze alloy are placed between the two flat surfaces of the part. Printed-in holes provide alignment between the two halves. The assembly is held together with stiff wire and brazed in a hydrogen braze furnace. Again, though this is a manual

process, it is considerably simpler than the brazing in conventional manufacturing since it only includes two complex parts with well-defined alignment features. Also, reducing the number of braze joints decreases the likelihood of leaks.

Two sets of ten test cavities (shown in Figure 7) were manufactured. These test cavities all had the same design geometry, identical to Cavity 2 of the full circuit. In addition to the test cavities, an existing X-band klystron circuit design currently in development at SLAC has been adapted for additive manufacturing. The circuit is designed to output 300 kW at 11.424 GHz. The final device shown in Figure 8 contains four cavities approximately 20 mm in diameter, the beampipe, input and output waveguides, mounting features for a collector and electron gun, as well as integrated tuning pins.

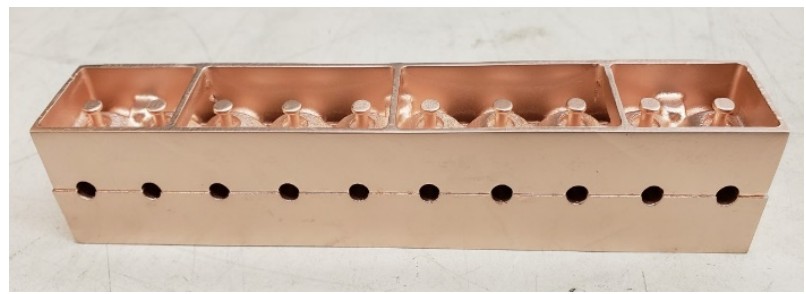

**Figure 7.** Complete brazed test block containing 10 cavities.

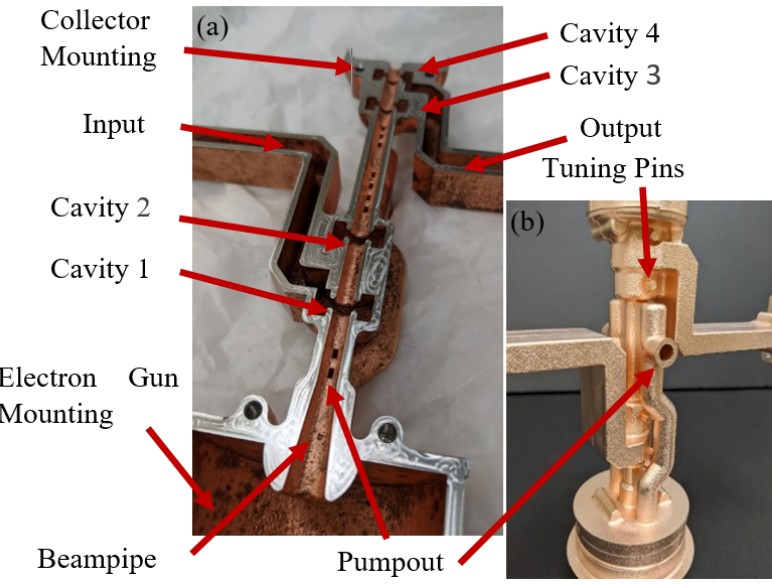

**Figure 8.** Diagram of features included in klystron circuit. (**a**) Unbrazed half-piece. (**b**) Full, brazed circuit.

## 4. Simulation and Characterization

An eigenmode simulation using ANSYS HFSS was performed to establish the expected cavity performance. The cavity was modeled with no external coupling to match the geometry used in the test cavities. The cavity cross-section and E-field is shown in Figure 9. The walls of the cavity were modeled as copper with no surface roughness. The expected values for cavity Q and frequency are tabulated in Table 1. Additionally, the entire klystron is simulated using KlyC to establish expected performance of the system.

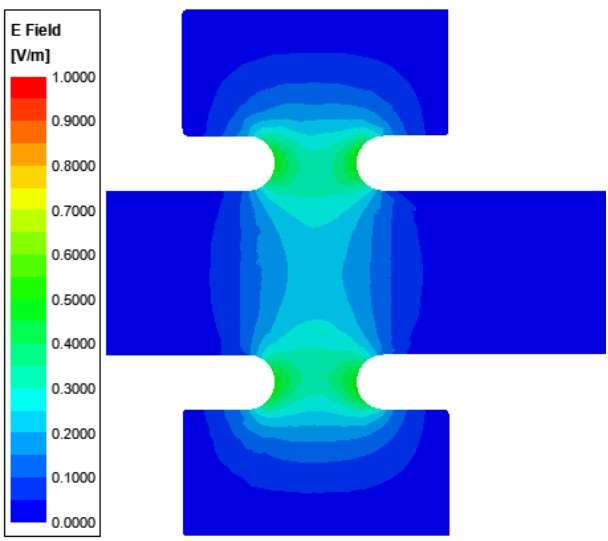

**Figure 9.** Normalized E-field of Cavity 3 simulated in HFSS.

**Table 1.** Cavity Resonance and Design Properties.

| Cavity | 1 | 2 | 3 | 4 |
|---|---|---|---|---|
| Frequency (GHz) | 11.424 | 11.44 | 11.495 | 11.41 |
| Quality Factor | 200 | 2000 | 2000 | 125 |

Cavities were measured using two E-field probes on an Agilent N5241A network analyzer. During measurements before brazing, the two halves were clamped together. Each probe was positioned near the irises of each successive cavity using manual slide stages. The peak resonance of each cavity was determined by inserting both probes completely into each cavity, then slowly removing them while monitoring the reflection and transmission coefficient curves. Once it was obvious that removing each probe further did not affect the resonant frequency (i.e., the cavities are not being detuned), a measurement was taken. Since the metal body of the probes extended through the entire beampipe, except in the cavity under test, other cavities are shorted. A waveguide load was placed on both the input and output waveguides during testing.

Every cavity of the full klystron circuit was measured both before and after brazing. The Q factor and frequency was calculated from the peak in the S12 transmission coefficient. To characterize the variation inherent in every manufacturing step, the 20 test cavities were characterized immediately after printing, after electropolishing, after plating, after face machining, and after brazing.

## 5. Test Cavity Results

The as-printed resonance for all 20 test cavities is shown in Figures 10 and 11. The printed geometry includes extra material on the mating faces which is later removed during face machining. Before removal, this material significantly affects cavity geometry and therefore frequency and Q factor. After face-machining, all test cavities should match Cavity 2. The simulated geometry shown in this figure includes this extra material to provide a consistent reference. There is a significant spread in both frequency and Q, however, observe that Q factor is uniformly lower, and frequency uniformly higher than the design intent. These trends, once characterized, would allow the designer to compensate by designing the cavities to have a higher Q factor and lower frequency than intended.

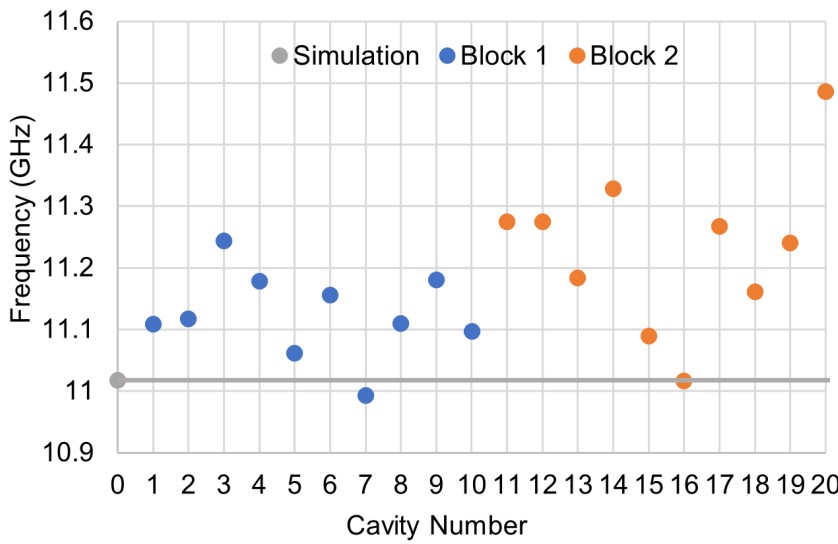

**Figure 10.** Frequency of all test cavities measured after printing. Simulated frequency is shown as horizontal grey line. Simulated geometry is adjusted to reflect cavity geometry before face machining.

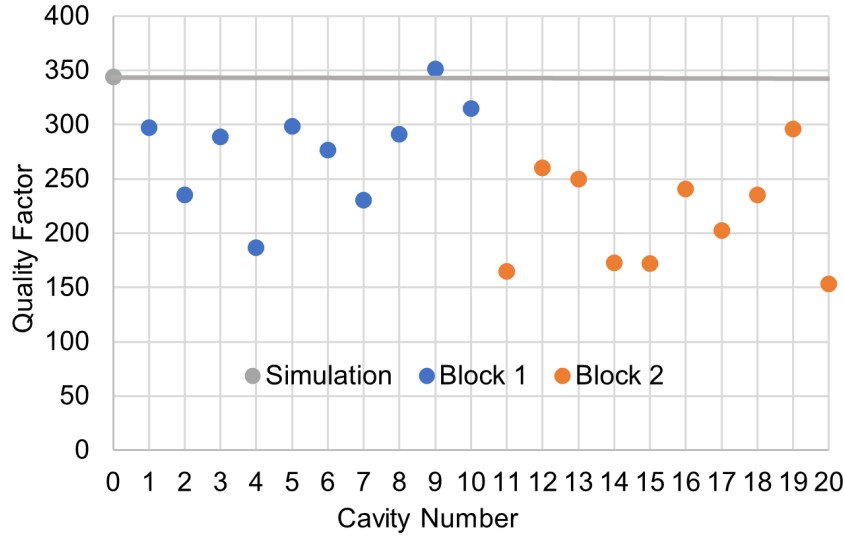

**Figure 11.** Q factor of all test cavities measured after printing. Simulated Q factor is shown as horizontal grey line. Simulated geometry is adjusted to reflect cavity geometry before face machining.

The cavity parameters before and after each major manufacturing step are shown in Table 2. After polishing, the frequency and frequency standard deviation rise. The Q also rises, but this is expected given that the lowered surface roughness would contribute to a lower effective surface resistance. After plating, Q rises significantly since the cavity surface is copper. Standard deviation for both frequency and Q decreases. After plating, half the cavities were face machined and measured while only clamped, and the other half was face machined, then brazed. After machining, the frequency and quality factor rises significantly as the cavity dimensions are brought to their final values. While not a direct comparison, the frequency seems to rise further after brazing, possibly because of shifts in the geometry during the braze. The quality factor appears to drop after brazing, possibly because of diffusion of the bulk steel into the Cu coating increasing the effective surface resistance.

**Table 2.** Test Cavity Properties Throughout Manufacturing Process.

|  | **As Printed** | **Polished** | **Plated** | **Face Machined *** | **Brazed *** |
|---|---|---|---|---|---|
| Mean Frequency (GHz) | 11.178 | 11.378 | 11.380 | 11.565 | 11.709 |
| Frequency σ (GHz) | 0.115 | 0.181 | 0.125 | 0.107 | 0.115 |
| Mean Q Factor | 246 | 313 | 1961 | 1317 | 1206 |
| Q Factor CV | 0.228 | 0.265 | 0.243 | 0.48 | 0.191 |

\* Data from different sets of 10 test cavities. Other data from all 20.

The final brazed cavity resonance of ten cavities is shown in Figure 12. There is significant variation in both frequency and Q factor, but the standard deviation of the frequency is only 115 MHz. During the printing process, the 3D model provided to the printer must be sliced into layers, then converted into a series of instructions for the printer to execute to create the part. Combined with phenomena inherent to the physical process of sintering powder, these factors mean that the actual printed part geometry is not identical to the model provided to the printer. Two fundamental types of variations are identified.

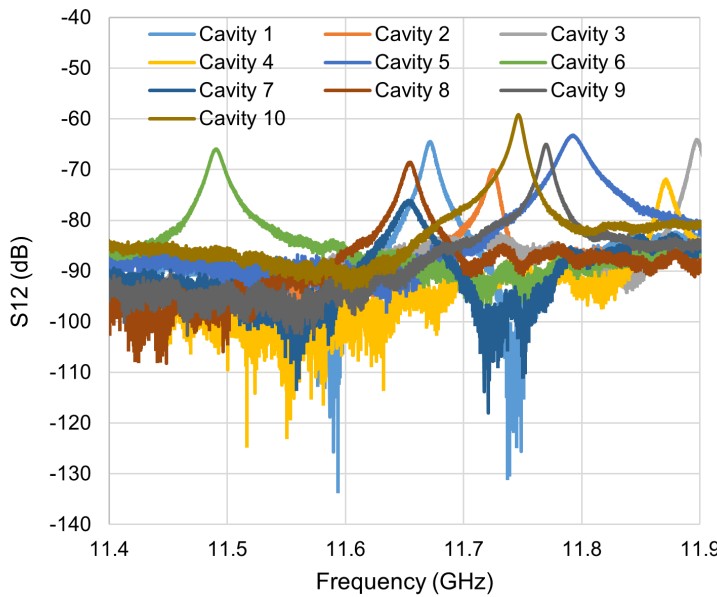

**Figure 12.** Transmission coefficient of 10 test cavities measured after brazing.

Firstly, there are consistent variations which similarly affect all parts. For example, it has been shown that small holes in DMLS parts are consistently printed smaller than the model [20]. These variation work to offset the mean frequency and mean quality factor from the theoretical values. Once these variations are characterized, the designer can compensate. For example, the mean frequency of the final cavities as shown in Table 2 is 285 MHz higher than the theoretical frequency. To compensate, the designer could target a frequency of 11.139 GHz. Assuming the offset remains consistent, the actual mean frequency of those cavities would be much closer to desired frequency of 11.424 GHz.

Secondly, inconsistent variations which may differently affect each theoretically identical part. These inconsistent errors contribute to the imprecision of the part and work to increase the standard deviation observed in the cavities' frequency and Q factor. Since these variations are inherently not predictable, they provide a more difficult challenge to

creating accurate cavities. Work must go into both limiting these variations and compensating for them via tuning. According to these results, achieving tuning on the order of the 115 MHz standard deviation of the final cavities would allow most cavities to be tuned to a precise frequncy.

After cold testing, the cavity surfaces were inspected with a laser confocal microscope. An image of the inner surface of a cavity with average roughness is shown in Figure 13. The surface roughness of 20 cavity surfaces was measured. The average area surface roughness (Sa) was 48 μm with a standard deviation of 12 μm.

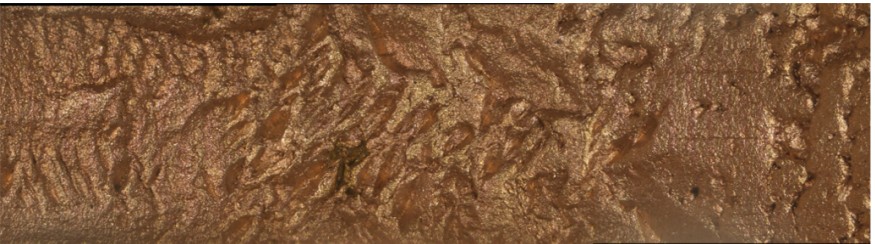

**Figure 13.** Microscope image of cavity surface.

## 6. Tuning Study

Tuning conventionally-manufactured copper cavities via mechanical deformation is relatively well-understood. However it is expected that the different mechanical properties of stainless steel, as well as the single-piece design of print-in-place tuning pins, will behave differently during tuning. To quantify this behavior, a preliminary tuning study was performed by impacting one of the completed, brazed test cavities multiple times. Between each impact, the cavity resonance was measured using the standard technique. The results of this study are shown in Figure 14. Through five rounds of tuning, the cavity frequency was raised more than 150 MHz. These traces also indicate that the Q factor was compromised.

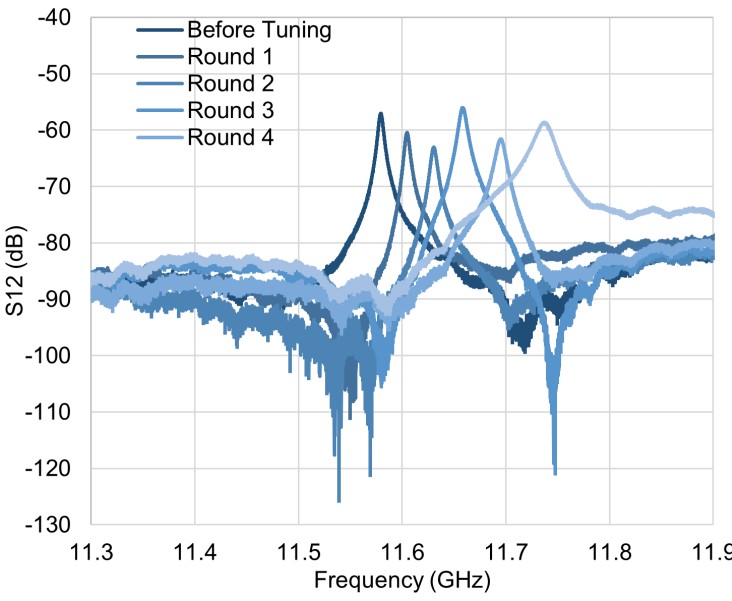

**Figure 14.** Transmission coefficient of test cavity measured after successive rounds of tuning.

Additionally, as can be seen in Figure 15, the tuning pin is significantly deformed. In a different tuning test, the pin broke completely through the cavity wall, as shown in Figure 16.

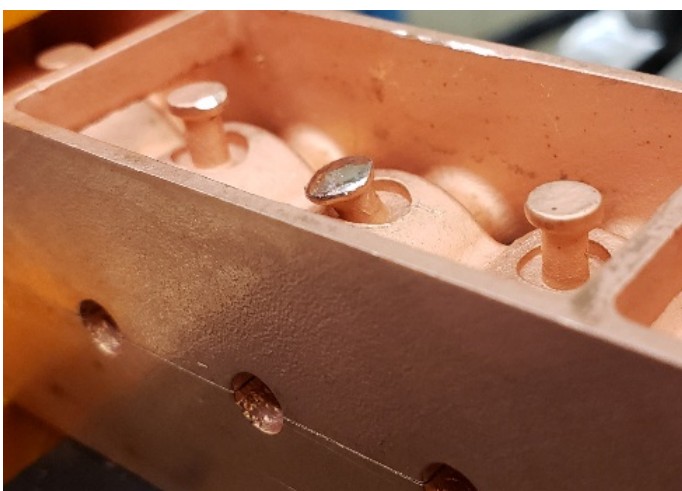

**Figure 15.** Deformed tuning pin after tuning test.

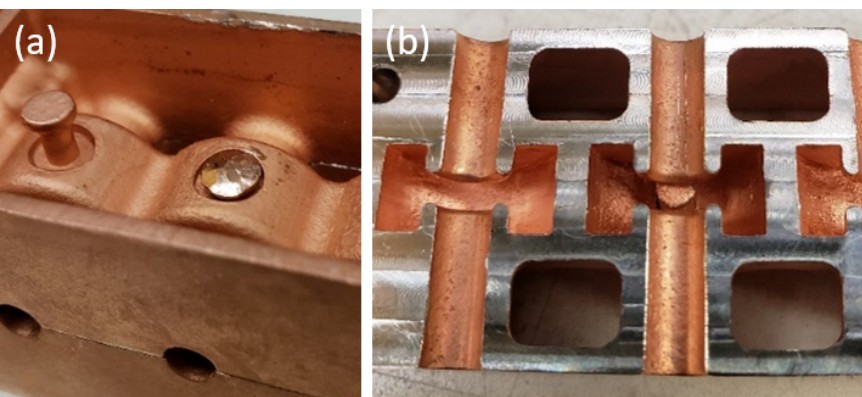

**Figure 16.** Damaged cavity after failed tuning test. (**a**) Tuning pin head on exterior of cavity. Undamaged pin is shown on the **left**. (**b**) Interior of damaged cavity with pin protruding. Undamaged cavity is shown on the **left**.

An additional tuning study was conducted on eight of the test cavities. The results are presented in Table 3. For this study, a tuning tool was fabricated. The tool grasps the head of each tuning pin and can smoothly push or pull the pin with a screw. Four cavities each were tuned upwards and downwards to the maximum extent permitted by the tuning tool. The resonance of each cavity was measured before and after tuning. After tuning, the cavities were cut open, and the cavity wall acted on by the tuning pin was measured on a laser confocal microscope. The deformation in the cavity wall caused by tuning was measured. These results establish an expected tuning bandwidth of 138 MHz with current tuning structures. As expected, the distance the cavity wall is deformed correlates to the frequency shift. Further, Q factor does not appear to be compromised through tuning, with some cavities even increasing in quality factor after tuning. Together, these results suggest that even with current tuning pin geometry, the 138 MHz tuning range is sufficient to cover the 115 MHz standard deviation of the as-fabricated cavity frequency, thus allowing most cavities to be brought to a precise frequency. Note that these tuning structures were completely unoptimized, and by affecting a larger portion of the cavity wall and therefore displacing more cavity volume, it is likely that significantly more effective tuning structures can be designed.

**Table 3.** Test Cavity Tuning Results.

| Cavity | Before | | After | | Frequency Change (MHz) | Deformation (μm) |
|---|---|---|---|---|---|---|
| | Frequency (GHz) | Q | Frequency (GHz) | Q | | |
| 1 | 11.671 | 1284 | 11.778 | 1220 | 107 | 516 |
| 2 | 11.897 | 1250 | 11.84 | 1275 | −57 | −347 |
| 3 | 11.725 | 1407 | 11.819 | 1195 | 94 | 511 |
| 4 | 11.871 | 1180 | 11.988 | 1095 | 117 | 438 |
| 5 | 11.608 | 1195 | 11.575 | 1197 | −33 | −297 |
| 6 | 11.49 | 967 | 11.457 | 962 | −33 | −320 |
| 7 | 11.583 | 566 | 12.029 | 1101 | 446 * | 648 |
| 8 | 11.655 | 906 | 11.619 | 1030 | −36 | −350 |

Average increase: 106 MHz, Average Decrease: −32 MHz. * Excluded from average due to questionably large shift.

## 7. Klystron Circuit Results

The klystron circuit underwent a helium leak check which showed a good braze joint with no leaks. Further, a DMLS sample was placed in vacuum to assess the material's suitability under vacuum. No significant outgassing was observed.

The measured resonant peaks for Circuit Cavities 3 and 4 are shown in Figure 17. Both cavities show significant resonance. Cavity 3 is very clear and shows relatively high Q. Cavity 4 is less clear, likely due to coupling into the connected output waveguide. Determining the peak on Cavity 4 was difficult due to the proximity to the end of the beampipe. The resonance of all circuit cavities both before and after brazing are shown in Figures 18 and 19. There is significant variation both between the cavities and between each cavity and its intended frequency. Cavity 2 exhibits the largest difference between intended and actual frequency at 4.6%.

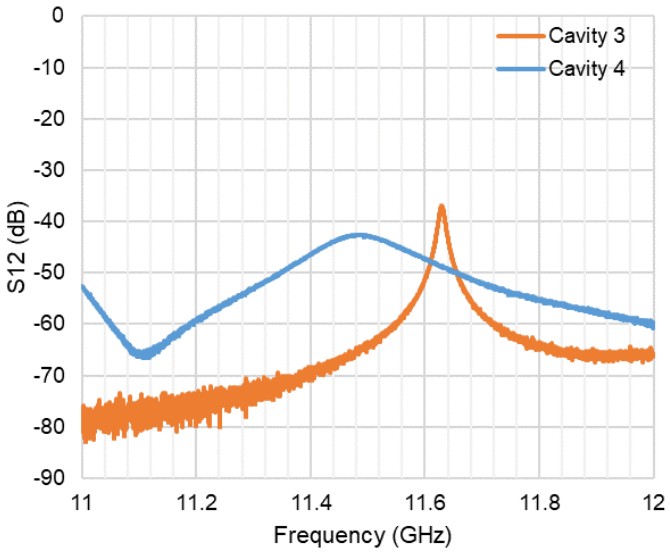

**Figure 17.** Measured transmission coefficients of cavities 3 and 4 measured after brazing.

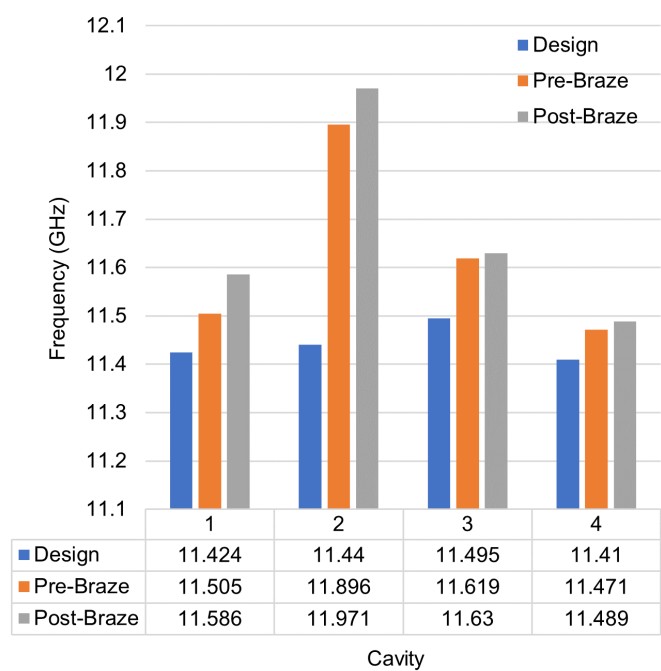

**Figure 18.** Resonant frequency of klystron circuit cavities as simulated, before brazing, and after brazing.

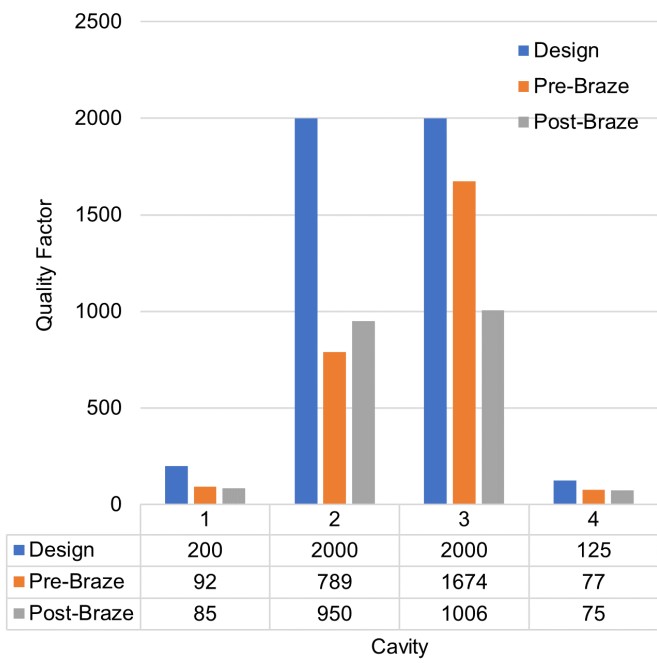

**Figure 19.** Quality factor of klystron circuit cavities as simulated, before brazing, and after brazing.

The measured Q factor shows significant promise. First, the final post-braze resonance is very nearly half of the design Q for every cavity. Second, with a Q factor of around 1000, Cavities 2 and 3 demonstrate that this manufacturing process can indeed create cavities with Q factors on the order necessary for klystrons.

To inform further testing and determine expected performance, the klystron was simulated, and an optimization was performed. Two potential tuning solutions are presented. In both cases, the operational frequency of the klystron has been changed from 11.424 GHz. Allowing flexibility in this parameter enabled considerably higher output power for this test device. In a production klystron, the designer could target a specific operational frequency

by offsetting the designed frequency of the cavities. In these simulations, it is assumed that tuning does not affect the Q factor.

Solution 1 is shown in Table 4. Due to the large frequency offset of Cavity 2, the rest of the cavities must be tuned by nearly 400 MHz, outside the established tuning range. This would require refined tuning structures which permit a larger tuning range. However, this solution allows an output power of 257 kW at a frequency of 11.955 GHz.

**Table 4.** Klystron Tuning Solution 1.

| Cavity | Frequency (GHz) | Tuning Needed (MHz) |
|--------|-----------------|----------------------|
| 1 | 11.955 | 369 |
| 2 | 11.971 | 0 |
| 3 | 12.000 | 370 |
| 4 | 11.941 | 542 |

Solution 2 is shown in Table 5. In this solution, Cavity 2 is untuned, so the large frequency offset means the cavity does not significantly participate in the klystron operation. Because of this, the remaining cavities need to be tuned significantly less, even less than the tuning range achievable with current tuning structures. However, effectively removing Cavity 2 reduces the power output to 55 kW at 11.620 GHz.

**Table 5.** Klystron Tuning Solution 2.

| Cavity | Frequency (GHz) | Tuning Needed (MHz) |
|--------|-----------------|----------------------|
| 1 | 11.620 | 34 |
| 2 | 11.971 | 0 |
| 3 | 11.630 | 0 |
| 4 | 11.620 | 131 |

## 8. Discussion

Although early in development, the measured RF properties and klystron simulations show promise that the manufactured klystron circuit can operate as a viable RF source. Additionally, the properties of the RF cavities indicate that only incremental progress is necessary to create consistent and accurate AM cavities as part of a tightly integrated and cost-effective device. However, further testing is necessary to prove vacuum tubes manufactured with this process can be functional.

The manufactured device should be pumped down to determine whether the structure can achieve a low enough base pressure. The slightly porous structure of the DMLS stainless steel as well as brazing in a hydrogen atmosphere may introduce additional outgassing and adsorption.

Due to the observed inconsistencies, further investigation and reduction of the large variations present in cavity frequency and quality is critical to the successes of this manufacturing process. These inconsistencies likely arise through multiple phenomena.

Firstly, the surface roughness introduces variation into the surface resistance of cavity walls, as well as small-scale geometry, which could impact RF behavior. Multiple routes for further development exist. DMLS was carried out by a commercial vendor and was likely optimized for acceptable and repeatable performance across many different metrics relevant to many different industries. Refining DMLS process parameters specifically for this application could yield lower-roughness parts more suitable for RF structures. Additionally, post-processing steps like electropolishing or bead blasting could be investigated in more detail to reduce roughness after printing.

Second, DMLS has been shown to introduce bulk geometry errors, such as small holes being smaller than expected. Further study should carefully measure the as-printed geometry and characterize these errors. It would be especially valuable to know if and how the designer could compensate for these errors before printing.

With only incremental improvement in the variation of RF properties and tuning bandwidth, X-band AM cavities can be cost-effectively produced to target a specific frequency and Q factor. AM provides the flexibility to quickly achieve this incremental improvement and integrate the resulting cavities into the next generation of klystrons and other high-power X-band devices.

**Author Contributions:** Conceptualization, C.W., J.M. and E.A.N.; methodology, C.W. and J.M.; validation, J.M. and E.A.N.; formal analysis, C.W. and B.S.; investigation, C.W., B.S. and G.M.; resources, E.A.N. and B.W.; data curation, C.W. and B.S.; writing—original draft preparation, C.W. and B.S.; writing—review and editing, C.W., B.S., E.A.N. and J.M.; visualization, C.W. and B.S.; supervision, E.A.N.; project administration, E.A.N.; funding acquisition, E.A.N. All authors have read and agreed to the published version of the manuscript.

**Funding:** This research has been supported by the U.S. Department of Energy (DOE) under Contract No. DE-AC02-76SF00515.

**Data Availability Statement:** The data presented in this study are available on request from the corresponding author.

**Acknowledgments:** The authors would like to thank Michael Collari and Keyence for providing the material characterization shown in Figure 6.

**Conflicts of Interest:** The authors declare no conflicts of interest.

## Abbreviations

The following abbreviations are used in this manuscript:

| | |
|---|---|
| RF | Radio Frequency |
| AM | Additive Manufacturing |
| FFF | Fused Filament Fabrication |
| SLA | Stereolithography |
| DMLS | Direct Metal Laser Sintering |

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
