# Peer review of "Radio Frequency Properties of a 3D Printed Klystron Circuit"

_instruments, doi:10.3390/instruments8010009_

Round 1

Reviewer 1 Report

Comments and Suggestions for Authors

The paper describes the 3D printing process for manufacturing RF cavities and the whole RF circuit for an X-band RF power amplifier, i.e. klystron.

The authors present, describe and show the results in a very clear way for all the steps in the fabrication process including measurements at low RF power characterization.

To my knowledge this is the very first attempt towards the fabrication of such devices and this initial study is very promising.

It is clear, as also stated by the authors, that that the roughness of the material, as printed, is not as good as with conventional CNC machining. It would help the reader and community in general to know what this value is, if it was ever possible to measure it, not only for the as-printed cavity made out of SS but only after the final stage of copper plating and before brazing.

Moreover, what is the mechanical tolerances which are possible to achieve with this fabrication process? is there an estimate?

Best regards. 

Author Response

Dear Reviewer,

Thank you for taking the time to read and review my manuscript. Your suggestions were quite helpful as I refined this manuscript. Below I’ve responded to each of your comments individually.

  • “It would help the reader and community in general to know what this value is, if it was ever possible to measure it, not only for the as-printed cavity made out of SS but only after the final stage of copper plating and before brazing.”

We were only able to measure the surface roughness of the cavities after copper plating and before brazing. I have included these data on page 10, line 220 of the revised manuscript. The typical surface roughness of DMLS parts is well characterized in literature, such as in my reference 21 (L. G. Gallant, A. Hsiao, and G. McSorley, “Benchmark physical and mechanical property chararacterization of 316L stainless steel DMLS prints,” Progress in Canadian Mechanical Engineering. Volume 4, 2021.)

  • “Moreover, what is the mechanical tolerances which are possible to achieve with this fabrication process? is there an estimate?”

These questions are well-explored in literature, such as in the previously mentioned reference. I felt that including how tolerances affect the RF properties of the printed cavities was more relevant to the topic of this paper. 

Thanks again for your suggestions and comments,

Charlotte Wehner

Reviewer 2 Report

Comments and Suggestions for Authors

The authors write an excellent paper on AM of klystrons. suggestions listed below. More information on post-processing, specifically the electropolishing and plating steps, measurements of surface roughness, and description of the ANSYS model.

Page 4 line 117 “With the use of 3D printing, arbitrarily thin walls can be added to the cavity surfaces allowing larger tuning ranges.” At some point there will be warping or possible porosity issues if the wall is too thin, recommend revising this sentence.

Page 5 line 133: “The thickness of the copper plating is not crucial, provided that it sufficiently exceeds the skin depth at the target frequency” Cu tends to diffuse into SS at high temperatures such as a braze or HIPing cycle, this could be an issue even with plating exceeding a skin depth as the interdiffusion would increase surface resistance within a skin depth of the surface. Was there any observation of this during the braze cycle? What thickness of Cu plating? Have you considered printing the klystron cavities out of Cu alloy?

Page 5 line 137: Do you have a measurement of surface roughness in as-printed and plated condition? This measurement should be included in the paper.

Page 6 line 154: How was surface roughness and material resistance modeled in the ANSYS simulation? This should be described as this effects Q of the system?

Page 7 line 165: was the ANSYS simulation done with waveguide loads on the input/output waveguide? Is this why the simulated Q is so low compared to the body cavities? The details of the ANSYS model and simulation parameters should be included.

Page 7 line 169: Please add the electropolishing procedure and the as-printed and electropolished surface roughness.

Page 8: which test cavities in Table 1 are equivalent to the cavities measured in Fig 10? I assume they are cavity 2, but this should be re-mentioned for clarity.

Page 8: RMS Surface roughness measurements for each cavity should be added to table 2

Page 8 line 180: In table 2, how were these measurements taken at various steps of only 2 sets of 10 cavities were produced (clamped together?)? Were some of the cavities measured prior to brazing the cavity halves together? If so, the presence of brazed of un-brazed cavity halves could also effect the Q factor? A column for brazed cavities should be added to this table for easy comparison

Author Response

Dear Reviewer,

Thank you for taking the time to read and review my manuscript. Your suggestions were quite helpful as I refined this manuscript. Below I’ve responded to each of your comments individually.

  • Page 4 line 117 “With the use of 3D printing, arbitrarily thin walls can be added to the cavity surfaces allowing larger tuning ranges.” At some point there will be warping or possible porosity issues if the wall is too thin, recommend revising this sentence.

I have edited this section for clarity and to better express that the flexibility of 3D printing enables optimized tuning structures

  • Page 5 line 133: “The thickness of the copper plating is not crucial, provided that it sufficiently exceeds the skin depth at the target frequency” Cu tends to diffuse into SS at high temperatures such as a braze or HIPing cycle, this could be an issue even with plating exceeding a skin depth as the interdiffusion would increase surface resistance within a skin depth of the surface. Was there any observation of this during the braze cycle? What thickness of Cu plating? Have you considered printing the klystron cavities out of Cu alloy?

I edited this section to include more information about the coating process and skin depth. I also added a figure which shows how the Cu does indeed appear to diffuse into the bulk material. With our measurements, we weren't able to directly measure the thickness of the plating after brazing, so we're not able to rule out that the surface resistance was reduced. We did consider directly printing the klystron from a Cu alloy, but we found that this material for DMLS is much less mature than SS 316 and offers worse tolerances, surface roughness, and printability in general.

  • Page 5 line 137: Do you have a measurement of surface roughness in as-printed and plated condition? This measurement should be included in the paper.

We did have measurements from after the plating, which I have included on page 10, line 220. Unfortunately did not measure the roughness as-plated, but this is well-characterized in literature, such as in my references 21 and 22 (L. G. Gallant, A. Hsiao, and G. McSorley, “Benchmark physical and mechanical property chararacterization of 316L stainless steel DMLS prints,” Progress in Canadian Mechanical Engineering. Volume 4, 2021.)

  • Page 6 line 154: How was surface roughness and material resistance modeled in the ANSYS simulation? This should be described as this effects Q of the system?
  • Page 7 line 165: was the ANSYS simulation done with waveguide loads on the input/output waveguide? Is this why the simulated Q is so low compared to the body cavities? The details of the ANSYS model and simulation parameters should be included.

The material was modeled as plain copper with no roughness, and the cavity was modeled in isolation without any coupling geometry. The Q and frequency is different because the model was changed to match the as-printed geometry before the face-machining step, where there is extra material. I have added these details about the simulation in this section.

  • Page 7 line 169: Please add the electropolishing procedure and the as-printed and electropolished surface roughness.

Unfortunately I wasn't able to find much information about the electropolishing procedure. It was performed according to standard procedure at our plating shop. I have indicated this in the section. We also unfortunately didn't gather roughness data at the time, though I see how it would be useful.

  • Page 8: which test cavities in Table 1 are equivalent to the cavities measured in Fig 10? I assume they are cavity 2, but this should be re-mentioned for clarity.

I've re-mentioned this and edited a few other sections for clarity about this point.

  • Page 8: RMS Surface roughness measurements for each cavity should be added to table 2

Unfortunately we only gathered roughness data of the 10 unbrazed cavities (20 unbrazed halves) after face machining. In hindsight this would be very useful data to know, but I have added all the data we have in page 10, line 220.

  • Page 8 line 180: In table 2, how were these measurements taken at various steps of only 2 sets of 10 cavities were produced (clamped together?)? Were some of the cavities measured prior to brazing the cavity halves together? If so, the presence of brazed of un-brazed cavity halves could also effect the Q factor? A column for brazed cavities should be added to this table for easy comparison

The cavities were clamped together and cold tested between each manufacturing steps. Half the cavities were only face machined, and half were face machined and brazed. I have added the data of the brazed cavities to table 2. I have also clarified this in the section, as well as added some relevant discussion.

Thanks again for your suggestions and comments,

Charlotte Wehner